# Green Tea (*Camellia sinensis*) Extract Induces p53-Mediated Cytotoxicity and Inhibits Migration of Breast Cancer Cells

**DOI:** 10.3390/foods10123154

**Published:** 2021-12-20

**Authors:** Ronimara A. Santos, Emmanuele D. S. Andrade, Mariana Monteiro, Eliane Fialho, Jerson L. Silva, Julio B. Daleprane, Danielly C. Ferraz da Costa

**Affiliations:** 1Laboratory for Studies of Interactions between Nutrition and Genetics, Department of Basic and Experimental Nutrition, Rio de Janeiro State University, Rio de Janeiro 20550-013, Brazil; ronimaras@gmail.com (R.A.S.); manu.dutra.pbi@gmail.com (E.D.S.A.); juliobd@gmail.com (J.B.D.); 2Laboratory of Functional Foods, Institute of Nutrition Josué de Castro, Federal University of Rio de Janeiro, Rio de Janeiro 21941-902, Brazil; mariana@nutricao.ufrj.br (M.M.); elianefialho@yahoo.com.br (E.F.); 3National Institute of Science and Technology for Structural Biology and Bioimaging, Federal University of Rio de Janeiro, Rio de Janeiro 21941-902, Brazil; jerson@bioqmed.ufrj.br

**Keywords:** polyphenols, catechins, antioxidant capacity, cell viability, cell cycle

## Abstract

Green tea (GT) has been shown to play an important role in cancer chemoprevention. However, the related molecular mechanisms need to be further explored, especially regarding the use of GT extract (GTE) from the food matrix. For this study, epigallocatechin gallate (EGCG) and epigallocatechin (EGC) were identified in GTE, representing 42 and 40% of the total polyphenols, respectively. MDA-MB-231 (p53-p.R280K mutant) and MCF-7 (wild-type p53) breast tumor cells and MCF-10A non-tumoral cells were exposed to GTE for 24–48 h and cell viability was assessed in the presence of p53 inhibitor pifithrin-α. GTE selectively targeted breast tumor cells without cytotoxic effect on non-tumoral cells and p53 inhibition led to an increase in viable cells, especially in MCF-7, suggesting the involvement of p53 in GTE-induced cytotoxicity. GTE was also effective in reducing MCF-7 and MDA-MD-231 cell migration by 30 and 50%, respectively. An increment in p53 and p21 expression stimulated by GTE was observed in MCF-7, and the opposite phenomenon was found in MDA-MB-231 cells, with a redistribution of mutant-p53 from the nucleus and no differences in p21 levels. All these findings provide insights into the action of GTE and support its anticarcinogenic potential on breast tumor cells.

## 1. Introduction

*Camellia sinensis* leaves provide one of the most popular manufactured drinks in the world, thus representing the second most consumed beverage after water. China is the largest tea producing country with an output of 1.9 million tons, accounting for more than 38 percent of the world total. Tea is widely consumed all over the countries, especially in China and India, and usually categorized into three types—green, black, and oolong—due to leaf fermentation degree and processing [1,2,3]. Green tea (GT) is produced by drying and steaming tea leaves to prevent fermentation and is considered a great source of natural polyphenols that accounts for 30 to 42% of the composition of the dry extract of the plant, depending on cultivation conditions. The major groups of these polyphenols belonging to the group of flavonoids are catechins (flavan-3-ol) and include epicatechin (EC), epicatechin gallate (ECG), epigallocatechin (EGC) and epigallocatechin gallate (EGCG). The presence of numerous hydroxyl groups in these molecules gives them strong antioxidant properties [4,5].

Available scientific studies indicate a significant impact of the diet and its natural phytochemicals on the prevalence of chronic non-communicable diseases [6]. Anti-inflammatory and antioxidant as well as chemopreventive activity are considered the most important hallmarks of the GT catechins, encouraging the inclusion of tea in the daily diet [7].

Evidence from case-control studies suggests that GT consumption may promote reduced incidence of breast cancer [8,9]. A recent umbrella review and meta-analysis of observational studies regarding tea consumption and risk of cancer showed that high consumption compared with low GT consumption significantly lowered the risk of breast cancer [10]. Further investigation is needed to provide evidence of the GT role in the overall risk of cancer [11].

Overall, the use of GT in cancer research is a promising strategy considering that recent studies expected 19.3 million new cases of cancer and 10 million deaths from cancer in 2020, including breast cancer. Female breast cancer was pointed as the leading cause of global cancer incidence in 2020, with an estimated 2.3 million new cases, representing 11.7% of all cancer cases [12]. Metastatic breast cancer is responsible for more than 90% of cancer-related deaths [13]. The etiology of breast cancer is multifactorial and in recent years, preclinical and clinical research provided growing evidence with respect to the protective effects of bioactive plant derived compounds on cancer-related biological pathways [14]. Phytochemicals may affect the carcinogenesis process from initiation to progression by improving detoxification, enhancing innate immune surveillance, promoting DNA repair, and inhibiting cell proliferation [15,16]. The main mechanisms of action of catechins include the modulation of cell-cycle related proteins, induction of apoptosis, modulation of transcription factors and important intracellular enzymes involved in migration, invasion, metastasis, inhibition of inflammatory factors, and modulation of membrane receptors [17,18,19].

Molecular mechanisms involved in the physiopathology of cancer are still limited, but a promising field in oncology research is tumor-suppressor proteins such as the p53 family. p53 is involved with a range of antitumor mechanisms, including induction of cell cycle arrest, senescence, and apoptosis, and reducing proliferative signaling [20]. Interestingly, the p53 family is considered a chemotherapeutic target of polyphenols, including GT catechins [21,22].

Most studies regarding the effects of polyphenols are commonly conducted by the administration of supplements or isolated compounds, and include limitations such as side effects related to high doses and poor bioavailability of catechins [23]. The presence of diverse bioactive compounds in GT matrix may increase the bioaccessibility, bioavailability, and efficacy of catechins, assuring an advantage over isolated compounds. Moreover, GT beverages are better tolerated when compared to capsules or bolus, due to liver toxicity promoted by free catechins [24]. It is important to notice that, although with very low frequency, toxicological studies show a hepatocellular pattern of liver injury with isolated EGCG intake amounts (~140–1000 mg/day). One of the recognized factors that can contribute to the hepatotoxic effects is the bolus dose provided by certain dosage forms such as capsules and tablets. These data encourage the inclusion of tea in the daily diet, instead of supplementation [25].

Regarding the recent evidence about GT effects on breast cancer [26], we hypothesized that a green tea extract (GTE) obtained from the food matrix could promote p53-mediated anticancer effects on human breast cancer cells. We found that GTE selectively targeted MDA-MB-231 and MCF-7 cells, thus reducing cell viability and migration without cytotoxic effect on non-tumoral cells, which suggests a high potential for application in carcinogenesis. Our data also show the involvement of the tumor suppressor p53 and p21 protein in GTE-mediated cytotoxicity.

## 2. Materials and Methods

### 2.1. Chemicals and Reagents

All reagents were analytical grade and water was obtained with a Milli-Q system from Millipore (Bedford, MA, USA). Pifithrin-α and Alamar Blue^®^ were purchased from Invitrogen (Carlsbad, CA, USA). Dimethyl sulfoxide (DMSO), trypsin, antibiotics (penicillin, streptomycin), fetal bovine serum, Mammary Epithelial Growth Supplement (MEGS), Dulbecco’s Modified Eagle’s Medium (DMEM) media were purchased from Thermo-Fisher Scientific (Saint Louis, MO, USA). Mitomycin-C, iron chloride hexahydrate and 2,4,6-tris(2-pyridyl)-S-triazine (TPTZ) were purchased from Sigma-Aldrich Chemical Co (Saint Louis, MO, USA). Anti-p53 (DO-1) and anti-GAPDH (0411) antibodies were purchase from Santa Cruz Biotechnology (Santa Cruz, CA, USA). Anti-p21 (12D1) antibody was purchased from Cell Signaling Technology (Danvers, MA, USA). Anti-β-actin (A1978) antibody was purchased from Sigma-Aldrich Chemical Co (Saint Louis, MO, USA). Catechin standards were purchased from Indofine Chemical Co. (Hillsborough, NJ, USA).

### 2.2. Green Tea Extract Polyphenols Quantification and Antioxidant Capacity

To investigate the effect of infusion time and temperature on polyphenols content and antioxidant capacity, a commercial brand of oven-roasted green tea, grown in two mountainous regions of the Brazilian coast, where the climate is similar to Japan’s and whose processing employs Japanese technology, was acquired in a local market in Rio de Janeiro, Brazil. Aqueous infusions were prepared in a proportion of 1 g:40 mL and submitted at 70, 75, 80, 85, 90, 95, and 100 °C during 5, 10, and 15 min. Total polyphenols content was determined according to the Folin–Ciocalteu assay [27] and results were expressed as mg of gallic acid equivalents (GAE) per liter (mg EAG/L). Antioxidant capacity was determined by the Ferric Reducing Ability Power (FRAP) assay as described previously [28,29]. (FRAP reagent was prepared by mixing 2 mL of 10 mM TPTZ solution in 6 M HCl with 2 mL of 20 mM ferric chloride solution and 20 mL of 300 mM acetate buffer (pH 3.6) and warmed to 37 °C prior to analysis. In a microplate, 20 μL of samples and 180 μL of the FRAP reagent were added, followed by reading at 595 nm (Biochrom^®^ Asys UVM340; Holliston, MA, USA) after 4 min of incubation. Results were quantified based on a standard curve of ferrous sulfate and were expressed as mmol Fe^2+^ per liter of extract (mmol Fe^2+^/L). Each extract was analyzed in triplicate.

### 2.3. Characterization of Green Tea Extract Polyphenols by HPLC

After determination of the best GTE extraction condition, a fresh GTE infusion (80 °C/5 min) was prepared and freeze-dried (LD3000 Freeze Drier; Terroni, SP, Brazil) in order to be applied in the cell culture. GTE was stored at −20 °C protected from light. The contents of catechin in GTE were determined by High-Performance Liquid Chromatography (HPLC) analysis. The HPLC system (Shimadzu^®^, Kyoto, Japan) included two LC-20AD pumps, automatic injector SIL-20AHT, diode array detector SPD-M20A, system controller CBM-20A, and degasser DGU-20A5. Chromatographic separation of catechins was achieved using a reverse-phase column C18 (5 μm, 250 mm × 4.6 mm, Kromasil^®^, Darmstadt, Germany). The mobile phase consisted of a gradient of 0.3% formic acid and 1% acetonitrile in water (eluent A) and 1% acetonitrile in methanol (eluent B), at a flow rate of 1.0 mL/min. Prior to injection, the column was equilibrated with 18.2% B. After sample injection, the solvent composition changed to 20.2% B in 1 min, 43.4% B in 18 min, and 85.9% B in 23 min, and kept constant until 30 min between injections, 10 min intervals were used to re-equilibrate the column with 18.2% B. The eluent was monitored by DAD at 210 and 280 nm. The injection volume was 10 μL. Identification of catechin (C), epicatechin (EC), epigallocatechin (EGC), epigallocatechin gallate (EGCG) and epicatechin gallate (ECG) was performed by comparison with retention time and absorption spectrum of the respective standard. Quantification was performed by external calibration. Data were acquired by LabSolutions software (Shimadzu Corporation^®^, Sydney, Australia, version 5.82, 2015).

### 2.4. Cell Culture

The human breast epithelial carcinoma cell lines MDA-MB-231 (mutated p53–p.R280K) and MCF-7 (wild-type p53) were obtained from the American Type Culture Collection (ATCC; Manassas, VA, USA). Cells were cultured in DMEM containing 4.5 g/L glucose supplemented with 10% fetal bovine serum and 1% of penicillin/streptomycin. Cells were maintained at 37 °C in a humidified atmosphere containing 5% CO_2_. The non-tumoral MCF-10A cells, obtained from the American Type Culture Collection (ATCC; Manassas, VA, USA) were cultured at the same conditions, plus adding MEGS supplementation.

### 2.5. Cell Treatment

Breast cancer cells were subcultured into 24- or 96-well plates and, upon adherence, supplementation of SFB culture medium was reduced to 2% for cell cycle synchronization for at least 12 h until reaching 60% confluency. In order to set the IC_50_ value, cells were exposed for 24 or 48 h to the following concentrations of GTE: 31.2, 62.5, 125, 250, 500, 750, and 1000 μg/mL. To be applied in culture, the GTE was diluted in complete DMEM medium and filtered in 0.22 μm filter. The use of this approach in breast cells is justified by previous studies which detected the presence of phenolic compounds derived from plant extract sources, including intact catechins, in different non-intestinal tissues [30].

### 2.6. Cell Viability

After treatment, cells were exposed to 10% Alamar’s reagent diluted in culture medium for 3 h and held at 37 °C in a humidified atmosphere containing 5% CO_2_. The plate was read in a spectrophotometer at 570 and 600 nm, and the data were expressed as % of viability in comparison to the control. A trypan blue exclusion test was also performed to access the effect of GTE on cells viability. Cells were treated as described before, washed with PBS, and resuspended with 100 μL of trypsin in 500 μL of 2% DMEM. An aliquot was stained with Trypan Blue dye (1:1) and the viable cells were immediately counted in Neubauer’s Chamber using an optical microscope. Results were expressed in % of viable cells. The same experiment was performed using a specific p53 inhibitor (pifithrin-α).

### 2.7. Selectivity Index

The degree of selectivity of GTE was expressed as previously reported [31], in accord to Equation (1).
(1)$$\mathrm{Selectivity}\;\mathrm{index}\;\left(\mathrm{SI}\right)=\frac{\mathrm{IC}50\;\mathrm{tumor}\;\mathrm{cells}}{\mathrm{IC}50\;\mathrm{MCF}-10A}$$

### 2.8. Wound-Healing Assay

To determine the effect of GTE on cell migration capability, wound-healing assay was performed as previously described [32]. Cells were cultured as described before until reaching 50–60% confluence. Cells monolayers were then washed with PBS and scratched with a sterile plastic p10 pipette tip. Wounds were made in triplicates. The peeled off cells were removed with two PBS washes and a fresh media containing different concentrations of GTE and a proliferation inhibitor, mitomycin-C (0.5 μg/mL), was added. Cells were treated with 48 h and images at zero and final time-points were acquired under a bright-field microscope, using a 10× objective. Wound width was measured using the ImageJ software version 1.43p (NIH, Bethesda, NY, USA).

### 2.9. Western Blotting

GTE-treated cells for 24 h were lysed and the protein concentration was determined using a Bio-Rad Protein assay (Bio-Rad Laboratories, Richmond, CA, USA). Proteins (50–100 µg) were resolved on 10% SDS-polyacrylamide gel and transferred into PVDF membrane. The membranes were incubated with blocking buffer (50 mM Tris, 200 mM NaC1, 0.2% Tween 20, and 5% BSA) for 2 h followed by the incubation with anti-p53, anti-p21, anti-GAPDH, and anti-β-actin antibodies (1:1000) overnight at 4 °C. The membranes were then washed with TBS-T buffer and incubated with secondary antibody conjugated with peroxidase for 1 h at room temperature. A Clarity Western ECL Substrate kit (Bio-Rad Laboratories, Richmond, CA, USA) was used for chemodetection. Levels of GAPDH and β-actin were used as an internal control to verify the loading of proteins in the gel. The densitometric quantification of bands was performed using Image Lab software (Bio-Rad Laboratories, Richmond, CA, USA).

### 2.10. Immunocytochemistry

Cells were grown at 70–80% confluence in 24-well plates on 13 mm glass coverslips and after cell cycle synchronization were treated with different concentrations of GTE for 24 h. Subsequently, cells were washed with PBS and fixed with a methanol:acetone solution (1:1) for 30 min. After further washing with PBS, cells were labeled with the anti-p53 primary antibody (1:200–2 h) followed by secondary Alexa Fluor 488 anti-mouse antibody (1:1000–1 h) and Hoechst 33,342 to nucleus identification in a dark chamber. Glass coverslips were mounted in a drop of glycerol for microscopical analysis. To obtain the images, a confocal laser scanning microscope was used (Leica TCS SPE, Wetzlar, Germany).

### 2.11. Statistical Analysis

Data were expressed as averages of at least three independent measurements ± standard deviation (SD). We confirmed the data for normal distribution and homoscedasticity of the variance using Shapiro–Wilk test, and then the groups were compared using one-way analysis of variance (ANOVA). Differences between pifithrin-α treated groups and control groups were evaluated using a Student’s *t*-test. Differences between GTE treated groups and control groups were evaluated using a Student’s *t*-test for p53 and p21 expression. In all cases, *p* value < 0.05 was accepted as statistically significant using GraphPad Prism Software v6.01 (San Diego, CA, USA).

## 3. Results

### 3.1. Time and Temperature Did Not Influence Polyphenol Content and Antioxidant Capacity of Green Tea Extract

To determine the best infusion condition to obtain a catechin-rich GTE, antioxidant capacity and polyphenol content of GT leaves were compared over a range of infusion times and temperature combinations. FRAP values ranged from 6.1 to 15.7 mmol Fe^2+^/L of infusion (Figure 1A) and total polyphenol content ranged between 1015 to 1823 mg GAE/L of infusion (Figure 1B). Our data demonstrated no impact of the combination of time and temperature in the content of phenolic compounds and antioxidant capacity. From these results, the previous studies from literature and with the aim of mimic the routine conditions of preparation of green tea for consumption by the population, the binomial 80 °C/5 min was chosen to perform the next experiments.

HPLC analyses of catechin content were also performed and, as expected, EGCG was the major catechin in GTE, representing 42% of total catechins, followed by EGC (40%), ECG (12%) and C (6%) (Table 1).

### 3.2. Green Tea Extract Reduces MDA-MB-231 and MCF-7 Cells Viability

To evaluate the anticancer potential of GTE, the breast cancer cells (MDA-MB-231 and MCF-7) and the non-tumoral human breast cell line (MCF-10A) were exposed to a range of GTE concentrations (31.2–1250 µg/mL) for 24 and 48 h and cell viability was accessed. As shown in Figure 2, Trypan Blue (Figure 2A,C) and Alamar Blue (Figure 2B,D) assays together demonstrated that GTE had a cytotoxic effect on both breast cancer cells. Treatment with GTE for 24 h displayed significant changes in cell viability, resulting in an IC_50_ value of 133 and 324 μg/mL for MDA-MB-231 and MCF-7 cells, respectively, considering Trypan Blue assay. No cytotoxic effect was demonstrated at the same concentrations on non-tumoral cells MCF-10A (Figure 2E). GTE targets breast cancer cells selectively, and it was shown that it can be two times more specific for MDA-MB-231 cells than MCF-7 (Table 2).

### 3.3. Green Tea Extract Inhibits Breast Cancer Cell Migration

Since migration capability has been implicated in the control of tumor progression, we investigated if GTE could suppress the migration of MDA-MB-231 and MCF-7 cancer cells using the wound healing assay while cells were exposed to mitomycin C to inhibit the proliferation. Our results showed that GTE reduced the migratory capability of both cells considering decreasing wound closure (Figure 3). MCF-7 is naturally less invasive since it formed tightly cohesive structures displaying robust cell–cell adhesions. For this reason, the MCF-7 monolayer was not completely closed in the negative control, even after incubation for 48 h. However, a 30% reduction in cell migration was observed at this condition when cells were exposed to 324 μg/mL of GTE (Figure 3A,B). A pronounced inhibition of migration was evidenced in the MDA-MB-231 cells, which form loosely cohesive grape-like or stellate structures that characterize its very aggressive tumor phenotype profile (Figure 3C,D). It was possible to observe the complete closure of the wound in the untreated cells already in 24 h, a phenomenon that did not occur in the conditions that received the addition of 253 μg/mL of GTE.

### 3.4. p53 and p21 Were Modulated by Green Tea Extract

A variety of phenolic compounds target p53-family proteins, which represents a promising strategy for tumor control. Therefore, we investigated p53 levels on GTE-treated tumor cells (Figure 4 and Figure 5). GTE exposition increased the expression of wild-type p53 protein on MCF-7 cells (Figure 4A–C). The opposite effect was observed in p53-mutant MDA-MB-231 cell (Figure 5A–C). From the results obtained, we aimed to evaluate the influence of p53 on cell viability, and a new assay using pifithrin-α, a specific p53 inhibitor that blocks the transcription of p53 responsive genes, was conducted. Our results showed that inhibition of p53 led to increased survival of both tumor lines treated with GTE, especially MCF-7, where an almost 50% increase in the viability could be observed on cells treated with 250 μg/mL of GTE (Figure 4F and Figure 5F). The cyclin-dependent kinase (CDK) inhibitor p21 is recognized as the major mediator of p53-dependent cell cycle arrest. Since p53 expression levels were modulated by GTE on MCF-7 and MDA-MB-231 breast cancer cells, we also evaluated GTE-effects on p21 expression. Our results demonstrated that 24 h GTE exposure at 324 μg/mL of MCF-7 cells promoted a significant activation of p21, whereas no effect was observed on MDA-MB-231 cells (Figure 4D,E and Figure 5D,E).

## 4. Discussion

Natural products are a source of anticancer bioactive compounds and attracted great interest in cancer research. GT polyphenols have demonstrated chemopreventive and chemotherapeutic properties, by inhibiting mutagenesis as well as tumor promotion and progression [16,33]. Despite the advances in research, carcinogenesis is a complex and multistage process and its underlying mechanisms are not yet well understood [34,35].

In the present study, we showed that the GTE obtained from the food matrix, rich in EGCG and EGC, was able to modulate p53 levels and reduce viability and migration on breast cancer cells. GT polyphenols are colorless, astringent, water-soluble compounds sensitive to different forms of degradation [36]. For this reason, the application of GT as an antioxidant source precedes the definition of the optimal conditions for the extraction of flavonoids. Thus, we first considered the effects of infusion time and temperature on the total polyphenols content of GT. Our findings demonstrate that the conditions adopted did not influence the total polyphenols content or antioxidant capacity of GTE.

However, it is correct to assume that high temperatures exert a greater influence on the degradation of the polyphenols since condition above 80 °C facilitates epimerization of catechins [37]. Aiming to mimic the process generally used in tea preparation by consumers, we choose for the binomial 80 °C/5 min. to the next analysis. We also determined the content of catechins of GTE by HPLC and the four mains catechins described in GT leaves with the predominance of EGCG and ECG were identified, supporting the potential for the application in cell culture. Studies have shown that Brazilian tea has a greater amount of phenolic compounds when compared to teas from other countries, due to characteristics of our climate, because the content of EGCG seems to be benefited by summer compared to spring [38].

Plant sources are widely used for naturally derived anticancer agents [39,40,41]. GT is a complex food matrix and tea catechins pharmacokinetics was previously studied. EGC, EC, and EGCG were detected on human plasma 1–2 h after drinking the equivalent of two cups of tea. Although the greatest fraction of EGCG ingested does not reach the bloodstream and EGC and EC seem to be more bioavailable, EGCG has a longer plasma half-life [42]. In rats, the isotope (-)-[4-3H] epigallocatechin gallate was detected in different organs and tissues, such as the liver, brain, eye, thymus, lung, heart, spleen, liver, pancreas, kidney, testis, prostate, and adrenal gland [30]. These findings demonstrate the absorption and delivery capacity of at least part of intact catechins to body tissues, thus suggesting a potential for biological application. Although intact catechins reach the bloodstream and were detected in a number of tissues, the fermentation by colonic microbiota is a crucial step in the metabolism of phenolic compounds that could not be analyzed in this study, which denotes a limitation.

The antitumor potential of catechins on different experimental models has previously been described in the literature. However, it is important to understand its effects as presented in the food matrix rather than the use of isolated catechins. It is known that exposure to these compounds as naturally occurring in food is an important factor for its antioxidant power and biological bioavailability, strengthening the importance of studies that fill this gap [14]. For this reason, GTE was applied in two breast tumor lines with different phenotypic characteristics. MDA-MB-231 cells showed a highly invasive and metastatic phenotype and expressed mutant p53 protein. MCF-7, less invasive cells, formed tightly cohesive structures displaying robust cell–cell adhesions and expressed wild-type p53 [43]. GTE demonstrates the potential to reduce the viability of both tumor cell lines, downregulating mitochondrial enzyme activity and promoting plasma membrane damage without, however, causing a cytotoxic effect on non-tumor cells (MCF-10A). In addition, the selective index (SI) of GTE to tumor cells demonstrates little potential for adverse effects, since an SI value less than two indicates general toxicity of the compound [31]. It is important to note that GTE doses above 500 μg/mL were toxic to all cells studied, reducing the population by more than 70%.

Cell invasion and migration play a key role in cancer progression and constitute a challenge for tumor treatment. In this study, we demonstrated that GTE reduces cell migration in both tumor cells, with a more pronounced effect on MDA-MB-231, an interesting finding taking the highly aggressive characteristics of this cell line. A possible explanation for how GT promotes this phenomenon operates on the capacity that phenolic and non-phenolic fractions of GT have to regulate several genes responsible for cell morphology, movement, and cytoskeletal formation. Seo et al. (2016) have demonstrated that 25 μg/mL of the GTE affected the molecular function and morphology of U2OS-GFP-α-tubulin cells, and the authors postulated that the mechanism of inhibition of cell migration is probably related to the rupture of the microtubules of the cytoskeleton [44]. Another hypothesis suggests that EGCG induces the expression of TIMP-3, a gene that negatively regulates matrix metalloproteinases, in both MDA-MB-231 and MCF-7 cell lines, thus culminating in the inactivation of metalloproteinases and containing tumor invasion [45]. GT catechins and polyphenols reduced five-fold the expression of MMP-9 in MDA-MB-231 cells [46].

To examine the possible molecular mechanisms by which GTE reduces cell viability, we investigated the involvement of the p53 tumor suppressor protein, a well-established target of polyphenols [47]. GTE increased wild-type p53 expression in MCF-7 cells and inhibition of p53 transcription genes by pifithrin-α seems to increase cell survival. This finding may provide clues as to how GTE reduces the tumor population, since increased p53 expression may lead to the induction of cell death. EGCG (20 and 40 µmol/L) led to the increase in Bax/Bcl-2 ration on PC-12 cells, which is the determining factor for apoptosis [48]. EGCG also reduced esophageal cancer cells ECa109 viability (concentrations from 0, 25, 50, and 100 to 200 mg/L) while significantly increasing the rate of apoptosis [49].

Alterations in the protein expression of cell cycle and apoptosis-related genes after exposition to GTE from the food matrix provide further clues about its mechanisms of action. In this study, we observed upregulation of wild-type p53 and a significant increase in p21 levels in MCF-7 cells. Previous studies indicated that EGCG treatment induced apoptosis in pancreatic cells due to the generation of ROS and caspase-3 and -9 activations, leading to cell-cycle arrest at G1 phase via controlling expressions of cyclin D1, cdk4, and p21CIP1 [18]. Employing nanotechnology, the effects of EGCG nanoparticles on expressions of several key regulatory proteins in PI3K-Akt pathway in MCF-7 cells was previously investigated and the treatment resulted in a significant increase in the protein expression of p21 and Bax [50]. Acting as a transcription factor, p53 activates p21 gene, which encodes p21 protein [51,52]. Thus, under the experimental conditions presented, the ability of p53 to induce its major molecular target appears functional in GTE-treated MCF-7. These data can explain, at least in part, the mechanisms involved on GTE effects on breast cancer MCF-7 cells.

p53 mutations, described in over 50% of all human cancers, lead not only to p53 loss-of-function, but also to negative-dominant effects (neutralization of the non-mutated p53) and gain-of-function effects, assuring a more aggressive phenotype, i.e., increasing metastatic capacity, genomic instability, and resistance to chemotherapy [53,54]. GTE decreased the mutant p53 p.R280K expression in MDA-MB-231 cells and immunocytochemistry data suggests a redistribution of mutant-p53 on the cellular nucleus. As expected for a mutant p53 tumor cell line, p21 levels did not change in MDA-MB-231 cells [52].

## 5. Conclusions

Our findings indicate that GTE obtained from the food matrix selectively targeted breast cancer cells, thus reducing cell viability and migration in MDA-MB-231 and MCF-7, which suggests a high potential for application in carcinogenesis. We also observed an increase in p53 and p21 proteins expression levels on MCF-7 cells, suggesting that this phenomenon may, at least in part, be mediated by p53 protein. The opposite effect was observed in MDA-MB-231 cells, which is interesting data, considering that p53 mutations were associated with gain of oncogenic function effects.

The main limitation of the present study is that we did not assess the exact mechanisms involved in the absorption and biotransformation of phytochemicals present in GTE by breast cancer cells. We believe that the mixture of bioactive compounds with a wide range of biological activities could have additive or synergistic effects against carcinogenesis.

This work provides new insights into the role of green tea as a whole food in p53-mediated cytotoxicity and inhibition of tumor cell migration. Future studies are planned to investigate the role of p53 in GTE-induced effects in vivo and 3D cancer cell cultures, in comparison to isolated catechins.

## Figures and Tables

**Figure 1 foods-10-03154-f001:**
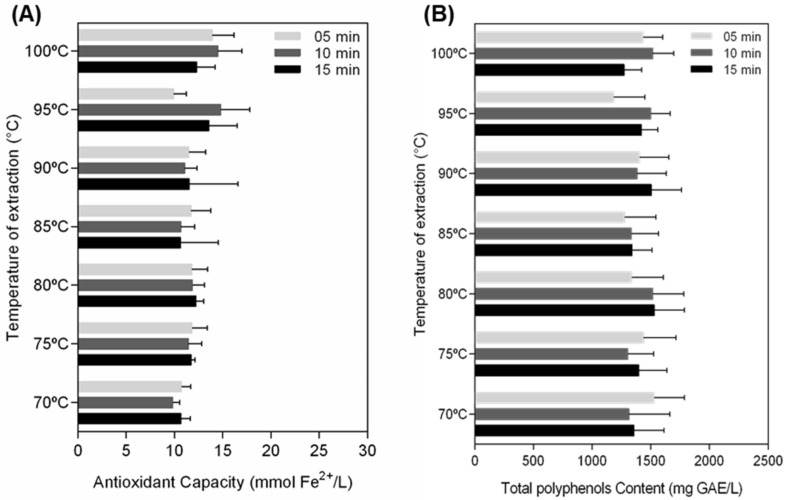
Effect of infusion time and temperature on antioxidant capacity and total polyphenols content of GTE. Antioxidant capacity (**A**) and total polyphenols content (**B**) were assessed by FRAP and Folin–Ciocalteau assays, respectively, after 5, 10, and 15 min of water extraction in different temperatures, as indicated. The results were expressed as averages of three independent measurements ± SD.

**Figure 2 foods-10-03154-f002:**
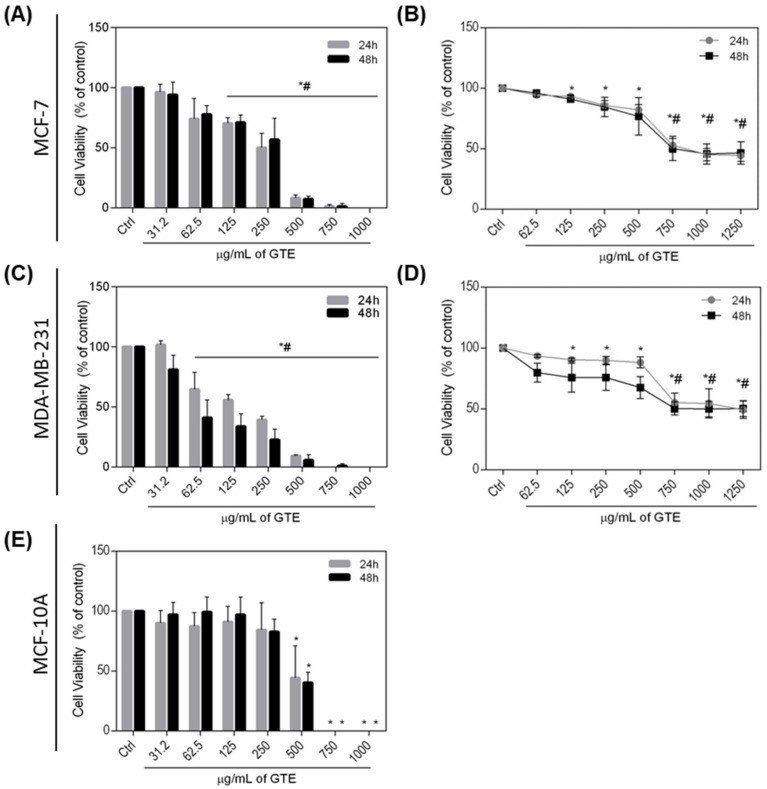
GTE treatment inhibits the proliferation potential and cell viability of human breast cancer MCF-7 and MDA-MB-231 cells. Cells were treated with different GTE concentrations for 24–48 h, as indicated. Cell viability was assessed by Alamar Blue (**A**,**C**) and Trypan Blue (**B**,**D**) assays. GTE was also tested on non-tumoral MCF-10A cells and viability was assessed by Alamar Blue assay (**E**). Experiments were performed in triplicate and results were expressed as % of controls. Symbols * and # indicate significant difference from controls (*p* < 0.05).

**Figure 3 foods-10-03154-f003:**
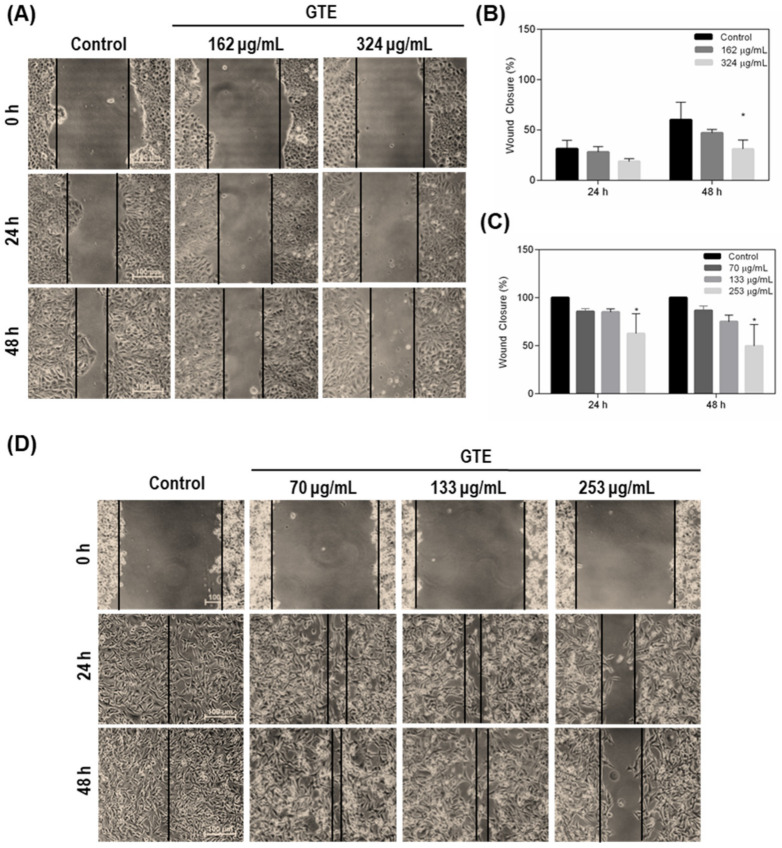
GTE reduces MCF-7 and MDA-MB-231 cells migration. Cell migration was assessed by hound healing assay during 24 and 48 h. Cells were cultured to 60% confluence, scratched with a sterile plastic tip and treated with GTE. Mitomycin C (0.5 μg/mL) was added as a cell proliferation inhibitor. The images of MCF-7 cells (**A**) and MDA-MB-231 cells (**D**) were acquired and the wound width was measured using the ImageJ software (*n* = 3) (**B**,**C**). We considered statistically different (*) those where *p* < 0.05.

**Figure 4 foods-10-03154-f004:**
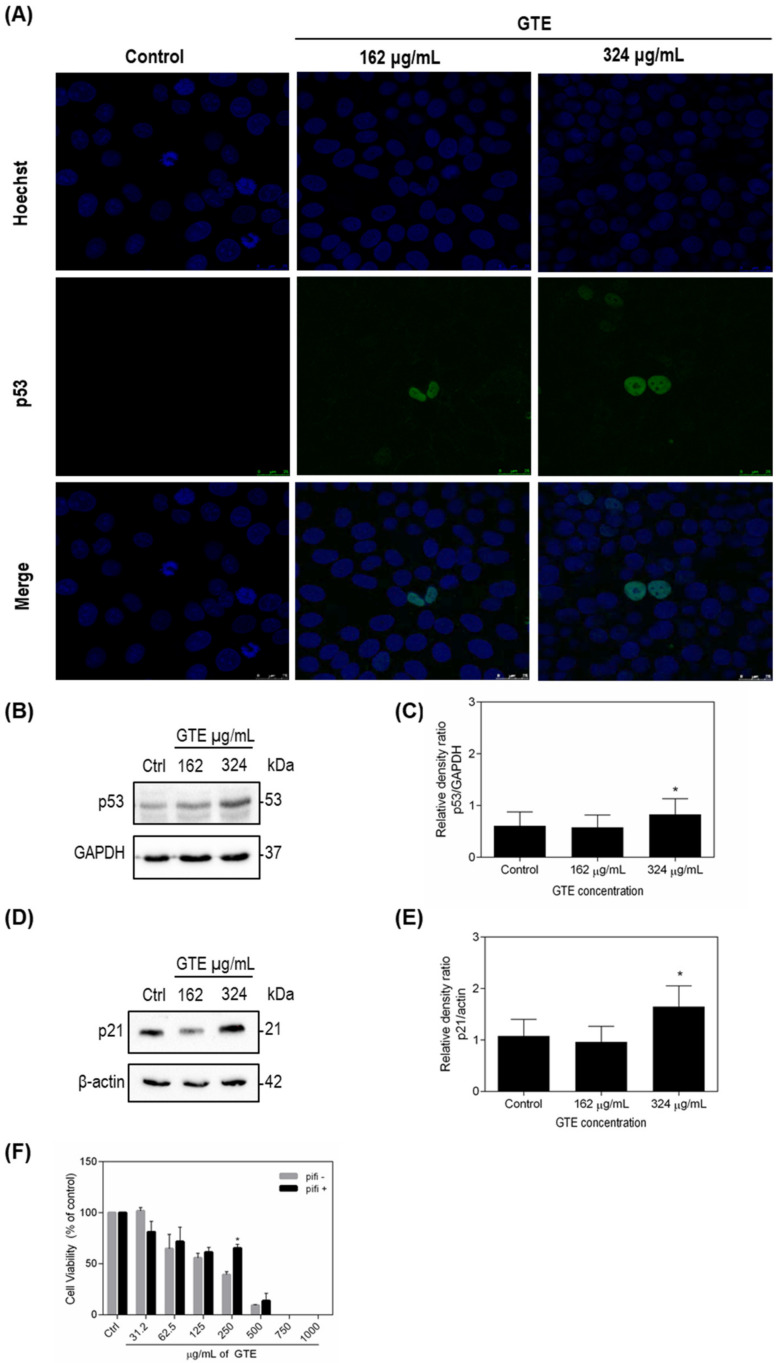
GTE stimulates p53 and p21 in MCF-7 cells. Immunocytochemistry assay demonstrated p53 distribution on cells (**A**). p53 and p21 protein levels were accessed by Western blotting and GAPDH or β-actin were used as controls. The densitometric quantification of bands shows p53 (*n* = 5) (**B**,**C**) and p21 (*n* = 3) (**D**,**E**) levels in MDA-MB-231. Cell viability of GTE-treated cells was assessed by Trypan Blue in the presence of a p53 inhibitor (pifithrin-α) and results were expressed as % of controls (*n* = 3) (**F**). Symbol * indicates significant difference from controls (*p* < 0.05).

**Figure 5 foods-10-03154-f005:**
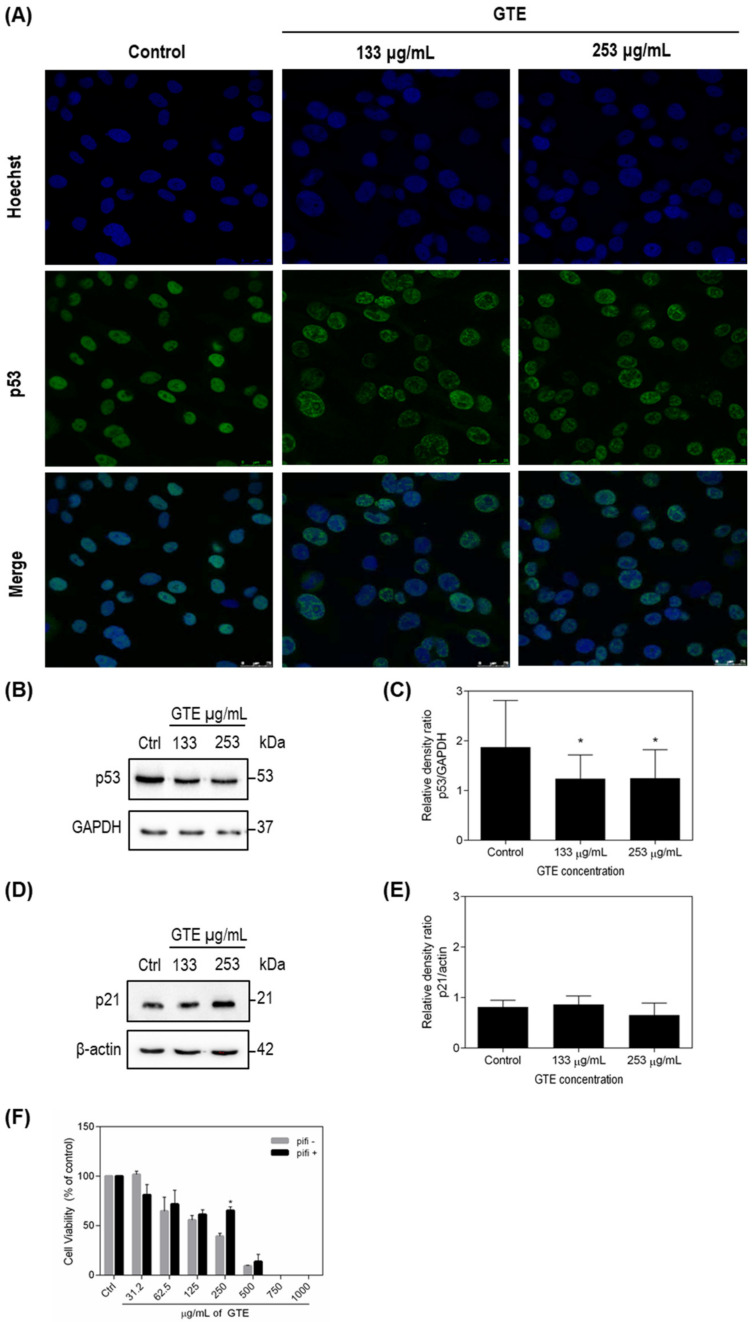
GTE stimulates p53 but not p21 in MDA-MB-231 cells. Immunocytochemistry assay demonstrated p53 distribution on cells (**A**). p53 and p21 protein levels were accessed by Western blotting and GAPDH or β-actin were used as controls. The densitometric quantification of bands shows p53 (*n* = 5) (**B**,**C**) and p21 (*n* = 3) (**D**,**E**) levels in MDA-MB-231. Cell viability of GTE-treated cells was assessed by Trypan Blue in the presence of a p53 inhibitor (pifithrin-α) and results were expressed as % of controls (*n* = 3) (**F**). Symbol * indicates significant difference from controls (*p* < 0.05).

**Table 1 foods-10-03154-t001:** Phenolic compounds contents in green tea extract.

Compound	Content (µg/mL)
Catechin (C)	5.06 ± 0.15
Epicatechin gallate (ECG)	9.44 ± 0.01
Epigallocatechin (EGC)	31.64 ± 0.69
Epigallocatechin gallate (EGCG)	32.83 ± 0.15
∑ catechins	78.97 ± 0.44

**Table 2 foods-10-03154-t002:** Comparison of cytotoxic activities of GTE.

IC_50_ (µg/mL)	MCF-7	MDA-MB-231
24 h	324	133
Selectively index (SI)	31.1	75.9

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
