# Peer review of "Green Tea (Camellia sinensis) Extract Induces p53-Mediated Cytotoxicity and Inhibits Migration of Breast Cancer Cells"

_foods, 2021, doi:10.3390/foods10123154_

Round 1

Reviewer 1 Report

I reviewed this manuscript, in the present study the authors show the effect of exposure to green tea extract (GTE) of breast cancer cells for 24 and 48 h, supporting its anticarcinogenic potential. In particular, GTE selectively targeted breast tumor cells without cytotoxic effect on non-tumor cells and p53 inhibition led to an increase in viable cells, especially in MCF-7, suggesting the involvement of p53 in GTE-induced cytotoxicity. GTE was also effective in reducing MCF-7 and MDA-MD-231 cell migration in 30% and 50%, respectively. An increment in p53 and p21 expression stimulated by GTE was observed in MCF-7, and an opposite phenomenon was found in MDA-MB-231 cells, with a redistribution of mutant-p53 from the nucleus and no differences in p21 levels.

The authors have provided a clear description of their study.

I provided my comments below.

    • Point 1.  Line 43: insert the reference number “Koch,2019”
    • Point 2.  It is possible to enrich paragraph 2.2 by adding a reference on the FRAP assay?
    • Point 3.  Line 149: change the verb form. The verb “was” does not seem to agree with the subject.
    • Point 4.  Line 155: specify the different concentrations of GTE that have been added.
    • Point 5.  The authors study breast cancer cell migration inhibited by p53, why did they not also evaluate the apoptotic effect of p53?
    • Point 6.  Line 390: please insert reference no. 38.

Author Response

Dear reviewer,

As requested, we enclose a revised version of the manuscript “Green tea (Camellia sinensis) extract induces p53-mediated cytotoxicity and inhibits migration of breast cancer cells”, which constructively addresses all of the concerns and suggestions of the Reviewers. The lines indicated in the responses below refer to the final version submitted with the accepted changes.

Point 1.  Line 43: insert the reference number “Koch, 2019”

Answer: As suggested, the reference was inserted in the text (line 46).

Point 2.  It is possible to enrich paragraph 2.2 by adding a reference on the FRAP assay?

Answer: The antioxidant capacity was determined by the Ferric Reducing Ability Power (FRAP) assay as established by Benzie & Strain in 1996. The same author has conducted a study aimed to compare in vitro antioxidant power of 25 different types of teas (Camellia sinensis) due FRAP assay tree years later (Benzie, 1999). These articles were used as references to our study. We added one more reference on 2.2 paragraph (line 127).

Point 3.  Line 149: change the verb form. The verb “was” does not seem to agree with the subject.

Answer: As required, the sentence has been corrected (line 164).

Point 4.  Line 155: specify the different concentrations of GTE that have been added.

Answer: In order to set the IC50 value, cells were exposed for 24 h or 48 h to the following concentrations of GTE: 31.2 μg/mL, 62.5 μg/mL, 125 μg/mL, 250 μg/mL, 500 μg/mL, 750 μg/mL and 1,000 μg/mL. We added this information in Materials and Methods section (lines 170-173).

Point 5.  The authors study breast cancer cell migration inhibited by p53, why did they not also evaluate the apoptotic effect of p53?

Answer: The cell viability experiments were assessed in the presence of pifithrin-α, a small molecule that inhibits p53 protein and prevents p53-mediated apoptosis. Our findings showed that inhibition of p53 by pifithrin-α led to increased survival of both tumor lines treated with GTE, especially MCF-7. For this reason, we believe that the reduction in cell viability promoted by GTE is, at least in part, due to p53-induced apoptosis. This phenomenon and other molecular pathways involved have been explored in ongoing studies by our group.

Point 6.  Line 390: please insert reference no. 38.

Answer: As suggested, the reference was inserted in the text (now reference 45, line 409).

Other minor modifications are indicated throughout the text with the Microsoft Word “Track Changes” function in the file. We thank you for the helpful suggestions and comments, which have greatly helped to improve our manuscript.

Sincerely yours,

Prof. Danielly C. Ferraz da Costa, Ph.D.

Laboratory for Studies of Interactions between Nutrition and Genetics

Institute of Nutrition – Rio de Janeiro State University 

São Francisco Xavier, 524, Pavilhão João Lyra Filho, 12º andar, 

Sala 12.150, Bloco F, 20550-013, Rio de Janeiro, RJ – Brazil 

Phone: +55 21 2334-1037

Reviewer 2 Report

Did the green tea samples have the same degree of crushing?

Author Response

Dear reviewer,

As requested, we enclose a revised version of the manuscript “Green tea (Camellia sinensis) extract induces p53-mediated cytotoxicity and inhibits migration of breast cancer cells”, which constructively addresses all of the concerns and suggestions of the Reviewers. The lines indicated in the responses below refer to the final version submitted with the accepted changes.

Did the green tea samples have the same degree of crushing?

Answer:  Our study started from a single tea sample, obtained from a commercial brand of oven-roasted green tea which presents the same degree of crushing. Green tea was grown in two mountainous regions of the Brazilian coast, where the climate is similar to Japan’s and whose processing employs Japanese technology. The leaves were firstly subjected to infusion for analysis of antioxidant capacity and content of total phenolic compounds. For application in cell culture, the infusion obtained was lyophilized and carefully homogenized. Aliquots were frozen at -20oC for the following tests.

Other minor modifications are indicated throughout the text with the Microsoft Word “Track Changes” function in the file. We thank you for the helpful suggestions and comments, which have greatly helped to improve our manuscript.

Sincerely yours,

Prof. Danielly C. Ferraz da Costa, Ph.D.

Laboratory for Studies of Interactions between Nutrition and Genetics

Institute of Nutrition – Rio de Janeiro State University 

São Francisco Xavier, 524, Pavilhão João Lyra Filho, 12º andar, 

Sala 12.150, Bloco F, 20550-013, Rio de Janeiro, RJ – Brazil 

Phone: +55 21 2334-1037

Reviewer 3 Report

Review of the article: Green tea (Camellia sinensis) extract induces p53-mediated cytotoxicity and inhibits migration of breast cancer cells

This study breast cancer MDA-MB-231 and MCF-7 cells and non-malignant breast epithelial cells (MCF-10A) were exposed to green tee extract.

The idea of the study is good. The paper is well written, but some editorial errors must be correct.  The conclusions consistent with the evidence and arguments presented.

Please correct some editorial errors, especially:

Line 229  Replace commas with dots in total polyphenol content.

Author Response

Dear reviewer,

As requested, we enclose a revised version of the manuscript “Green tea (Camellia sinensis) extract induces p53-mediated cytotoxicity and inhibits migration of breast cancer cells”, which constructively addresses all of the concerns and suggestions of the Reviewers. The lines indicated in the responses below refer to the final version submitted with the accepted changes.

Line 229. Replace commas with dots in total polyphenol content.

Answer: Total polyphenol content ranged between 1,015 to 1,823 mg GAE/L of infusion. Here we use the comma as a thousand separator, in accordance with the English language standard.

Other minor modifications are indicated throughout the text with the Microsoft Word “Track Changes” function in the file. We thank you for the helpful suggestions and comments, which have greatly helped to improve our manuscript.

Sincerely yours,

Prof. Danielly C. Ferraz da Costa, Ph.D.

Laboratory for Studies of Interactions between Nutrition and Genetics

Institute of Nutrition – Rio de Janeiro State University 

São Francisco Xavier, 524, Pavilhão João Lyra Filho, 12º andar, 

Sala 12.150, Bloco F, 20550-013, Rio de Janeiro, RJ – Brazil 

Phone: +55 21 2334-1037

Reviewer 4 Report

The article by Ronimara A. Santos et al. analyses the cytotoxic effect of Green tea (Camellia sinensis) extract on breast cancer cells of the MDA-MB-231 mutant line and compares these effects on a normal epithelial line to corroborate the impact on cancer. 
The article is well written, the methodology and results clearly reflect the conduct of the study and the conclusions are interesting, although human trials are needed to analyse the real impact of the tea extract on breast cancer prevention.
Some comments are attached to improve the readability of the introduction as well as the visibility of the article in the scientific community.

To increase the visibility of the article in bibliographic databases, it would be advisable to select keywords that do not appear in the title.

Introduction

  • Line 47-51. Are there any prospective observational or intervention studies in which this effect is observed? It would be advisable to include more references.
  • Line 52-54. Please incorporate epidemiological information on breast cancer 
  • Line 80. Repetitive sentence and similar to line 47

Author Response

Dear reviewer,

As requested, we enclose a revised version of the manuscript “Green tea (Camellia sinensis) extract induces p53-mediated cytotoxicity and inhibits migration of breast cancer cells”, which constructively addresses all of the concerns and suggestions of the Reviewers. The lines indicated in the responses below refer to the final version submitted with the accepted changes.

To increase the visibility of the article in bibliographic databases, it would be advisable to select keywords that do not appear in the title.

Answer: Thank you for the suggestion. We choose different keywords, as follows: polyphenols, catechins, antioxidant capacity, cell viability, cell cycle (lines 28-29).

Introduction. Line 47-51. Are there any prospective observational or intervention studies in which this effect is observed? It would be advisable to include more references.

Answer: The most recent umbrella review and meta-analysis of observational studies regarding tea consumption and risk of cancer showed that high consumption compared with low green tea consumption significantly lowered the risk of breast cancer, despite the authors reinforce the necessity of more well-designed prospective studies in this scenario (Kim, 2021). These data were added in the introduction section (lines 51-54).

Line 52-54. Please incorporate epidemiological information on breast cancer.

Answer: The commentary was considered, thus we added breast cancer data and update the epidemiological information in accord to Global Cancer Statistics 2020, as follow: “Overall, the use of GT in cancer research is a promising strategy considering that recent studies expected 19.3 million new cases of cancer and 10 million deaths from cancer in 2020, including breast cancer. Female breast cancer was pointed as the leading cause of global cancer incidence in 2020, with an estimated 2.3 million new cases, representing 11.7% of all cancer cases” (lines 59-62).

Line 80. Repetitive sentence and similar to line 47

Answer: We rewrite the sentence, as follows: “Regarding the recent evidences about GT effects on breast cancer, we hypothesized that a green tea extract (GTE) obtained from the food matrix could promote p53-mediated anticancer effects on human breast cancer cells” (lines 92-95).

Other minor modifications are indicated throughout the text with the Microsoft Word “Track Changes” function in the file. We thank you for the helpful suggestions and comments, which have greatly helped to improve our manuscript.

Sincerely yours,

Prof. Danielly C. Ferraz da Costa, Ph.D.

Laboratory for Studies of Interactions between Nutrition and Genetics

Institute of Nutrition – Rio de Janeiro State University 

São Francisco Xavier, 524, Pavilhão João Lyra Filho, 12º andar, 

Sala 12.150, Bloco F, 20550-013, Rio de Janeiro, RJ – Brazil 

Phone: +55 21 2334-1037

Reviewer 5 Report

Santos et al. gave us a study about green tea extract induced p53-meidated cytotoxicity and inhibited migration of breast cancer cells. I think it is not so excellent because it lacks of novelty about the study material green tea. There are so many studies about this object reported in all of the world especially in Asian countries, such as China, Japan and Korea. There are some critical issues should be clarified as follows:

  1. In the introduction, the detail information of Camellia sinensis leaves should be supplied, including the output per year, place of production, and mainly consumed country.
  2. The introduction of breast cancer should be more detail, about the mainly caused reason and main mechanism of cancer death.
  3. And the relative reported journals about GT inhibited cancer should be introduced in the part of introduction too.
  4. Usually, the contents of GT extracts are especially in phytochemicals and there are so many chemicals should be determined by more precise equipment.
  5. The concentration of GTE were only two for p53 and p21 expression in cells. Why only two? But your cell viability detection concentration were seven. And the concentration didn’t include the two in p53 and p21 expression.
  6. All data should supply the standard deviation.
  7. In figure 3, the concentration were chosen also not consistent with each other.
  8. In figure 4, the cells amount were too low in the Immunocytochemistry assay for p53.

Author Response

Dear reviewer,

As requested, we enclose a revised version of the manuscript “Green tea (Camellia sinensis) extract induces p53-mediated cytotoxicity and inhibits migration of breast cancer cells”, which constructively addresses all of the concerns and suggestions of the Reviewers. The lines indicated in the responses below refer to the final version submitted with the accepted changes.

  1. In the introduction, the detail information of Camellia sinensis leaves should be supplied, including the output per year, place of production, and mainly consumed country.

Answer: As requested, we added the following information at the beginning of the introduction section: “China is the largest tea producing country with an output of 1.9 million tonnes, accounting for more than 38 percent of the world total (FAO, 2015).  Tea is widely consumed all over the countries, especially in China and India, and usually categorized into three types – green, black, and oolong – due to leaf fermentation degree and processing” (lines 33-36).

  1. The introduction of breast cancer should be more detail, about the mainly caused reason and main mechanism of cancer death.

Answer: As suggested, additional information about breast cancer has been inserted and updated in the introduction section (lines 50-66):

 “Evidence from case-control studies suggests that GT consumption may promote a reduced incidence of breast cancer [8,9]. A recent umbrella review and meta-analysis of observational studies regarding tea consumption and risk of cancer showed that high consumption compared with low GT consumption significantly lowered the risk of breast cancer [10]. Further investigation is needed to provide evidence of the GT role in the overall risk of cancer [11].

Overall, the use of GT in cancer research is a promising strategy considering that re-cent studies expected 19.3 million new cases of cancer and 10 million deaths from cancer in 2020, including breast cancer. Female breast cancer was pointed as the leading cause of global cancer incidence in 2020, with an estimated 2.3 million new cases, representing 11.7% of all cancer cases [12]. Metastatic breast cancer is responsible for more than 90% cancer-related deaths [13]. The etiology of breast cancer is multifactorial and in recent years, pre-clinical and clinical research provided growing evidence with respect to the protective effects of bioactive plant-derived compounds on cancer-related biological pathways [14].”

  1. And the relative reported journals about GT inhibited cancer should be introduced in the part of introduction too.

Answer: As suggested, the following references were inserted in the introduction section (line 73):

Shirakami, Y.; Shimizu, M. Possible mechanisms of green tea and its constituents against cancer. Molecules 2018, 23, 2284, doi:10.3390/molecules23092284.

Claudia Musial, Alicja Kuban-Jankowska, M.G.-P. Beneficial properties of green tea. Int. J. Mol. Sci. 2020, 21, 1744.

  1. Usually, the contents of GT extracts are especially in phytochemicals and there are so many chemicals should be determined by more precise equipment.

Answer: We appreciate your consideration and we intend to use more accurate techniques for such experiments in ongoing studies by our group. For this study, the High-Performance Liquid Chromatography (HPLC) system was the only available equipment we had for the catechins identification.

  1. The concentration of GTE was only two for p53 and p21 expression in cells. Why only two? But your cell viability detection concentration was seven. And the concentration didn’t include the two in p53 and p21 expression.

Answer: To better understand the effects of GTE on breast cancer cells, a board of concentrations was initially tested in 24h and 48h. To analyze p53 and p21 expression, we used as reference the GTE IC50 values (24h) for MCF-7 (324 μg/mL) and MDA-MB-231 (133 μg/mL), including an additional lower or higher concentration (162 μg/mL for MCF-7, and 253 μg/mL for MDA-MB-231) to be tested.

  1. All data should supply the standard deviation.

Answer: Data were expressed as averages of at least three independent measurements ± standard deviation (SD). This information was included in the Material and Methods section (lines 233-234).

  1. In figure 3, the concentration were chosen also not consistent with each other.

Answer: To perform cell migration assays, we used as reference the GTE IC50 values (24h) for MCF-7 (324 μg/mL) and MDA-MB-231 (133 μg/mL), including one or two additional concentrations (162 μg/mL for MCF-7, and 70 μg/mL and 253 μg/mL for MDA-MB-231) to be tested. Such concentrations were established due to the different sensitivity of tumor cell lines to GTE.

  1. In figure 4, the cells amount were too low in the Immunocytochemistry assay for p53.

Answer: The number of cells used in the immunocytochemistry assays was established from previous studies by our group (Journal of Biological Chemistry, v. 287, n. 33, p. 28152-28162, 2012; Oncotarget, v. 9, n. 49, p. 29112, 2018; Protein Misfolding Diseases. Humana Press, New York, NY, 2019. p. 265-277). The nucleus staining with Hoechst 33342 in MCF-7 (Fig. 4A) reveals a number of cells similar to the experiments performed with MDA-MB-231 (Fig.5A). However, the weak staining intensity for p53 at the same conditions occurs because MCF-7 cells express the wild form of the protein, that does not accumulate in the nucleus like mutant-p53 in MDA-MB-231 cells.

Other minor modifications are indicated throughout the text with the Microsoft Word “Track Changes” function in the file. We thank you for the helpful suggestions and comments, which have greatly helped to improve our manuscript.

Sincerely yours,

Prof. Danielly C. Ferraz da Costa, Ph.D.

Laboratory for Studies of Interactions between Nutrition and Genetics

Institute of Nutrition – Rio de Janeiro State University 

São Francisco Xavier, 524, Pavilhão João Lyra Filho, 12º andar, 

Sala 12.150, Bloco F, 20550-013, Rio de Janeiro, RJ – Brazil 

Phone: +55 21 2334-1037

Round 2

Reviewer 5 Report

I think authors gave me a pretty good response to my quesiton and suggestion. It could be acceptable for publication in Foods.

Author Response

Dear reviewer, 

We thank you for the helpful suggestions and comments, which have greatly helped to improve our manuscript.

Sincerely yours,

Prof. Danielly C. Ferraz da Costa, Ph.D.

Laboratory for Studies of Interactions between Nutrition and Genetics

Institute of Nutrition – Rio de Janeiro State University 

São Francisco Xavier, 524, Pavilhão João Lyra Filho, 12º andar, 

Sala 12.150, Bloco F, 20550-013, Rio de Janeiro, RJ – Brazil 

Phone: +55 21 2334-1037
